# Cost-Effectiveness Analysis of the Helicobacter Pylori Screening Programme in an Asymptomatic Population in China

**DOI:** 10.3390/ijerph19169986

**Published:** 2022-08-13

**Authors:** Tianyu Feng, Zhou Zheng, Jiaying Xu, Peng Cao, Shang Gao, Xihe Yu

**Affiliations:** School of Public Health, Jilin University, Changchun 130012, China

**Keywords:** cost-effectiveness, gastric cancer, helicobacter pylori, Markov model, screening

## Abstract

Objective The aim of this study was to investigate the cost-effectiveness of Helicobacter pylori (*H. pylori*) screening and eradication treatment in an asymptomatic population in China and to explore the most cost-effective screening protocol for *H. pylori*. Method We used TreeAge 2019 to construct Markov models to assess the direct healthcare costs and quality-adjusted life years (QALYs) and the cost per year of life saved (YoLS) of three therapies, i.e., annual, triennial and five-yearly *H. pylori* screening. Excess probabilities were derived from published high quality studies and Meta-analyses, and costs and utilities were derived from the Chinese Yearbook of Health Care Statistics and published studies. Incremental cost-effectiveness ratios (ICERs) were used to describe the results. The willingness-to-pay threshold was set at China’s Gross National Product per capita. Result In the asymptomatic population, the ICER per QALYs gained was US$1238.47 and US$1163.71 for every three and five years of screening compared to the annual screening group; the ICER per YoLS gained was US$3067.91 and US$1602.78, respectively. Conclusion Screening for *H. pylori* in asymptomatic populations in China and eradicating treatment for those who test positive is cost-effective. Increasing screening participation in asymptomatic populations is more effective than increasing the frequency of screening. From a national payer perspective, it is cost-effective to screen the general asymptomatic population in China for *H. pylori* and to eradicate those who test positive. Individuals need to choose a screening programme that they can afford according to their financial situation.

## 1. Introduction

Helicobacter pylori (*H. pylori*) is a Gram-negative microaerobic bacterium that can be transmitted from person to person and is a major causative agent of gastric diseases such as chronic gastritis, peptic ulcers and gastric cancer [1,2]. China has a high prevalence of *H. pylori* infection and also has the highest number of peptic ulcers and gastric cancer cases [3]. The annual public health burden of *H. pylori*-related disease in China is enormous. The global consensus on *H. pylori* recommends that all people with *H. pylori* infection should receive eradication treatment unless there are clear circumstances that make eradication inappropriate [4].

The prevalence of *H. pylori* infection is closely related to regional economic development and the level of public health [5,6]. With the development of public health in recent years, the prevalence of *H. pylori* infection in China is decreasing year by year but the overall infection rate is still high [7,8]. Currently, the focus of the Chinese healthcare system is not on *H. pylori* infection, and even for those at high risk of *H. pylori*-related disease, there is no clear consensus strategy for the treatment of all infected individuals in China [9]. It is a widely accepted view in China that screening and treatment programmes for *H. pylori* in people presenting with peptic ulcer, gastric MALT lymphoma, chronic gastritis with dyspepsia and other *H. pylori*-related diseases should be implemented. However, the implementation of screening and treatment programmes for *H. pylori* infection in asymptomatic populations in China is controversial [9]. The main reason for this is the huge economic cost of aggressively screening and treating all *H. pylori*-positive patients and the ecological side effects of antibiotic use due to the very high prevalence of *H. pylori* in China [10].

Screening for *H. pylori* has been shown to be cost-effective in previous Markov model studies [11,12,13]. The study assumes annual screening for *H. pylori* in asymptomatic people. High frequency screening of asymptomatic people may not be a reasonable screening strategy [12]. There is a possibility of re-infection with *H. pylori* after cure, but the probability of re-infection is low. The prevailing view is that the population should be divided according to the risk of developing gastric cancer and recommendations for screening every 5 years, every 3 years or annually should be developed [14,15]. Repeated screening for *H. pylori* in asymptomatic populations is therefore indicated, but the frequency of screening and eradication treatment required to achieve the best cost-effectiveness needs to be further investigated. In addition, as *H. pylori* infections are more frequent in low-income groups, they are often unwilling to bear the costs associated with repeated screening [16]. It is therefore necessary to identify a more cost-effective screening strategy.

This study used a Markov decision model to evaluate the cost-effectiveness of different *H. pylori* screening and eradication strategies for the prevention of *H. pylori*-related disease in asymptomatic populations in China. The aim was to find the most cost-effective screening and treatment strategies.

## 2. Materials and Methods

### 2.1. Model Design

This study used a Markov model to conduct a cost-effectiveness analysis to assess the economic benefits of an *H. pylori* screening programme in China at an incremental cost per quality-adjusted life year (QALY) and the cost per year of life saved (YoLS). A Markov decision model was developed to evaluate three *H. pylori* screening regimens i.e., annually, every three years and every five years. Screening is carried out by the urea breath test (UBT) and those who screen positive receive 14 days of bismuth quadruple therapy. The model includes the development and treatment status of two diseases, peptic ulcer, and gastric cancer, to assess the cost-effectiveness of different screening strategies for *H. pylori*-related disease. The end of the model depends on the life expectancy of the Chinese population (78 years).

Figure 1 shows the Markov model and Table 1 shows the Parameters in the model.

### 2.2. Model Population

The study assumed a cohort population of 10,000 people and set their initial age at 20 years, with a cohort maximum year of 80 years (the average life expectancy of the Chinese population in 2020). It is assumed that they are initially healthy individuals, free of *H. pylori* infection and without associated digestive disorders.

### 2.3. Transition Probabilities

All variables used in our Markov models were derived from published studies (identified using PubMed and the Chinese National Knowledge Infrastructure) or authorities (China National Bureau of Statistics), as shown in Table 1. The transfer probabilities in the model are assumed to obey the Bate distribution. The 5-year survival rate for gastric cancer is reported to be 45–49.5% according to studies [20]. We transformed the 5-year survival rate of gastric cancer based on the declining exponential approximation of life expectancy (DEALE) principle [31,32], and the transformed equation was as follows.
(1)$$r=-\frac{1}{t}\ln\left(S\right)$$
(2)$$P=1-e^{-r*T}$$

In Equation (1), *S* is the five-year survival rate of gastric cancer, *t* is time, and *P* in Equation (2) is the probability of metastasis after transformation (Table 1).

In this study, a 14-day bismuth quadruple therapy was chosen as the *H. pylori* eradication treatment regimen, which selected a proton pump inhibitor, bismuth and two antibiotics (e.g., metronidazole and tetracycline). The therapy is recommended by the Chinese *H. pylori* guidelines and the report of the 5th Maastricht Conference [2,9]. The eradication rate of 14-day bismuth quadruple therapy was 85.51% (74.71–96.41%) according to published studies [19]. Mortality rates for non-gastric cancers were obtained from the China Health Statistics Yearbook 2020 for all ages [33].

### 2.4. Treatment Costs and Health Utility

In this study, parameters regarding the cost of treatment were obtained from reports published by the National Health and Family Planning Commission or the State Food and Drug Administration of China. The cost of bismuth in quadruple therapy is the market retail price of the drug. We assumed that people with no *H. pylori* infection and asymptomatic *H. pylori* infection were in a healthy state (health effect value of 1). The health utility values for PUD and gastric cancer are from a study conducted in 2017 [34]. In the baseline analysis, the QALY is discounted at 3% per annum.

The total cost is accumulated by multiplying the cohort size with the sum of the costs for each health state. The QALYs for each cycle are calculated using the utility value associated with each health state multiplied by the proportion of years lived in that state. Each cycle of the model is one year in length, so the model runs for a total of 60 cycles. The total QALY and life year is the accumulation of the QALY and life year values over all cycles.

### 2.5. Model Outcome

Notably, the following is according to the recommendation of the World Health Organization (WHO) for the evaluation of pharmacoeconomic [35]. ICER < 1 fold of gross domestic product (GDP) per capita, the increased cost is completely worth it and very cost-effective; 1 fold of GDP per capita < ICER < 3 fold of GDP per capita, the increased cost is acceptable and cost-effective; ICER >3 fold of GDP per capita, the increased cost is not worth it and not cost-effective. China’s GDP per capita in 2020 (US$11,000) is used as the threshold for willingness-to-pay (WTP). Costs of treatment are discounted at an annual rate of 5% according to the Chinese Pharmacoeconomic Assessment Guidelines [36]. We convert the fees to US dollars at the 2021 exchange rate (1 US$ is approximately 6 RMB)

### 2.6. Uncertainty and Scenario Analysis

Many of the parameters used in this study have considerable uncertainty. For this reason, one-way sensitivity analysis and probabilistic sensitivity analysis were conducted to assess the effect of uncertainty on the robustness of the results

The impact of each input parameter on the results was assessed by a deterministic sensitivity analysis (DSA) with a one-way sensitivity analysis (±20% of the input parameter), using tornado plots to present the results for each parameter.

Probabilistic sensitivity analysis (PSA) was used to determine the uncertainty of the input parameters. PSA was performed using Monte Carlo simulation simulations with 1000 iterations. Each parameter was specified a certain distribution, where the mean of the distribution is typically equal to the point estimate.

By varying the assumptions for scenario analysis, this study examines the cost-effectiveness of *H. pylori* screening in two scenarios, as follows.

It is assumed that screening for *H. pylori* is initiated at different ages in asymptomatic people. This hypothesis is used to explore the optimal screening regimen for different age groups.Compare the cost effectiveness of no repeat screening with that of no screening. This scenario examines the cost-effectiveness of increasing screening participation in asymptomatic people who are not screened.

The cost-effectiveness acceptability curve (CEAC) and the incremental cost and incremental quality adjusted life year cost-effectiveness (CE) plane are used to present the results of the probabilistic sensitivity analysis.

## 3. Result

### 3.1. Cost-Effectiveness Analysis

Table 2 reports the incidence of PUD, gastric cancer and death over the lifetime of the three *H. pylori* screening regimens in people not infected with *H. pylori*. There was no significant difference in the number of gastric cancer cases between the three cohorts; the difference in the incidence of PUD between the three groups was also not significant.

Baseline data are reported in Table 3. In the cohort with annual *H. pylori* screening, subjects survived an average of 45.43 years at an average total cost of $2487.08. In the cohort screened for *H. pylori* every three years, subjects survived an average of 45.01 years at an average total cost of US$2266.06. In the cohort screened for *H. pylori* every five years, subjects spent an average of 45.01 years at an average total cost of US$2241.19. Compared to the annual *H. pylori* screening regimen, the ICERs per QALY gained were US$1317.48 and US$1277.92, and the ICERs per YoLS gained were US$516.41 and US$522.05 for subjects using the every-three-year and every-five-year screening regimens, respectively. There was no significant benefit in terms of total QALYs, or number of gastric cancers, for subjects using an annual *H. pylori* screening programme compared to those screened several years apart. The annual screening programme for *H. pylori* did not show a significant cost benefit over the other two screening programmes.

### 3.2. Sensitivity Analyses

The results of the one-way sensitivity analysis are presented using a tornado diagram. The impact of extreme variations in each key parameter on ICER for different screening and treatment strategies is shown in the tornado diagram. In the Markov model, the cost-effectiveness of screening and treatment programmes is not sensitive to any of the variables (Figure 2). From the results it can be seen that the PUD-related parameters have the greatest influence on the results.

Figure 3 shows the cost effect curves for the three screening treatment strategies. The probability of the CEAC being cost-effective at a WTP of $113,000 was 89.80%, 8.70% and 0.40% for annual, triennial and five-yearly screening respectively. At a WTP level of $1500 or more per QALY, a treatment strategy with annual screening is >50% likely to be cost-effective (Figure 3). Programmes with triennial and quinquennial screening have a higher probability of being cost-effective when the willingness to pay is below $1500.

Figure 4A,B show CE plane of PSA results based on 1000 Monte Carlo simulations. The scatter is predominantly in the third quadrant, indicating that adopting a screening programme every three years or every five years costs less money but achieves relatively less QALYs than a strategy of screening treatment every year. The PSA results were similar to those of the basic analysis: an annual screening programme at a WTP of $11,000 was more cost effective.

### 3.3. Scenario Analyses

The results of the scenario analysis are shown in Table 4. Increasing the frequency of screening is more cost effective for older asymptomatic people.

The older the age of screening initiation, the more advantageous the annual screening programme is compared to other screening programmes. When the cost of screening is halved, the annual screening strategy is more cost effective over the whole time.

The results in Table 4 show a definite advantage of even a one-time *H. pylori* screening regimen over a no-screening regimen. In the CE plane, the scatter is concentrated in the fourth quadrant, meaning that subjects in the no-repeat screening strategy gained more QALYs and spent less compared to the no-screening cohort (Figure 5).

## 4. Discussion

This study provides new evidence for the development of *H. pylori* screening programmes in asymptomatic populations and assists Chinese health policy makers and digestive disease specialists in developing more cost-effective screening programmes and recommendations. Markov models are applicable to chronic disease research and effectively bridge the gap between real-world clinical observations. The use of Markov models to study the cost-effectiveness of different *H. pylori* screening programmes effectively fills a gap that is difficult to achieve in the real world. Our study used Markov models to conduct a health economic assessment of *H. pylori* screening and treatment programmes in an asymptomatic population and to determine the cost-effectiveness of different *H. pylori* screening programmes for the asymptomatic Chinese population.

### 4.1. Control of Helicobacter Pylori Infection

In the 2018 study, it was noted that the prevalence of *H. pylori* in China is very high [8]. *H. pylori* infection is therefore a very serious public health problem in China. The need for interventions to reduce the prevalence of *H. pylori* infection in China is very urgent for the public health sector in China. In the results of this study, we found that screening for *H. pylori* and eradication of patients who screened positive had a significant effect on improving the health and economic benefits of the asymptomatic population. The comparison cost $218.96 less per QALY obtained for *H. pylori* screening than for subjects who were not screened for *H. pylori*. This is consistent with the findings of previous studies. It is well known that China is currently facing a very serious threat of antibiotic resistance, which is also the main reason for the failure of *H. pylori* eradication treatment [37]. The rigorous use of antibiotics not only facilitates *H. pylori* eradication treatment, but also affects the effectiveness of antibiotics for other diseases [38]. For this reason we use bismuth quadruple therapy in our modelling to reduce the impact of this. The antibiotics used in this treatment, such as tetracycline and metronidazole, are less likely to be resistant in the treatment of *H. pylori* and also prevent patients from developing resistance to other antibiotics commonly used in clinical practice [2].

### 4.2. Effectiveness in the Prevention of Peptic Ulcers

Previous Markov model studies have compared the cost-effectiveness of *H. pylori* eradication therapy with PPI therapy alone in terms of short-term efficacy and lifelong prevention. The incidence and recurrence of PUD after *H. pylori* eradication was significantly reduced and more cost-effective than PPI treatment alone [39,40]. In this study we compared the cost-effectiveness of different screening programmes for the prevention of peptic ulcers and the results showed that there was little difference in the effectiveness of the three screening programmes. From the DSA results, the parameters regarding PUD had the greatest impact on the final outcome. This suggests that the improved health benefit of *H. pylori* screening in asymptomatic people comes mainly from the prevention of PUD. A programme of screening at intervals of several years can reduce expenditure compared to a programme of annual screening. Past studies have shown that reinfection after successful *H. pylori* eradication therapy is rare, so the benefits of eradication therapy for infected individuals are long-term [21]. In contrast, the results of the Markov model suggest that the health benefits obtained from screening for *H. pylori* in the asymptomatic population for high-stakes testing are not significant. High frequency screening for *H. pylori* is therefore of limited effectiveness in preventing PUD in patients who have already been screened and successfully eradicated.

### 4.3. Preventive Effect on Gastric Cancer

There is a strong association between *H. pylori* infection and the development of gastric cancer. According to a study by the Cancer Research Centre of Sun Yat-sen University, there are about 450,000 new cases of gastric cancer in China each year, of which at least 64.84% are caused by *H. pylori* infection [41]. The consensus report on the management of *H. pylori* infection in China recommends screening for *H. pylori* in communities at high risk of gastric cancer or in individuals at high risk of gastric cancer in China and the eradication of *H. pylori* in screen-positive patients. In previous Markov model studies, it has been shown that annual screening is more cost effective than no screening [42]. However, such a high frequency of screening tends to create problems of over-detection and unnecessary fear [12,13]. In contrast, in this study, we found that the difference in gastric cancer incidence between the cohorts of the three screening protocols was non-significant. Subjects in the screening group were able to obtain more QALYs at less expense in the cohort without repeat *H. pylori* screening compared to the cohort without *H. pylori* screening.

### 4.4. Screening Strategy Recommendations

China has a huge number of people infected with *H. pylori*. Large-scale screening for *H. pylori* in asymptomatic populations entails significant economic and labour costs [9]. And with successful eradication treatment, patients can reap lifelong health benefits [39,40]. Based on the results of this study, an annual screening programme is cost-effective from the perspective of national payers. However, from an individual payer’s perspective, the individual’s income and willingness to pay need to be taken into account to recommend a screening programme. In light of previous studies, we believe that a screening programme can be adopted for asymptomatic people on a five-year or longer cycle. Lowering the age at which screening is initiated can increase the health benefits for asymptomatic people. The ICER results suggest that different screening programmes should be adopted for asymptomatic people in different age groups. In conjunction with the findings of the Chinese Consensus on the management of *H. pylori* [9], we recommend that the screening interval for the general asymptomatic population should be set at every five years or longer. The frequency of screening can be increased for those who are older (>50 years), have digestive symptoms, or have a family history of gastric cancer.

In terms of screening frequency, increasing the frequency of *H. pylori* screening has a very limited effect on the prevention of *H. pylori*-related disease. The triennial and quinquennial screening programmes cost $1238.47 and $1163.70 more per QALY gained compared to the annual screening programme, respectively. The incremental QALYs and reduction in *H. pylori*-associated disease obtained with the annual screening programme did not show an absolute advantage over the triennial and quinquennial screening programmes. However, the benefit that the cohort without repeat screening could obtain compared to the group without screening was highly significant. We therefore believe that it is more cost-effective to increase the participation rate of unscreened asymptomatic people in screening than to increase the frequency of screening.

## 5. Strengths and Limitations

The strength of this study is that our study design includes two major diseases caused by *H. pylori* infection that can be prevented by screening, unlike most previous studies that were limited to the primary outcome of gastric cancer. The design of this study is closer to the real world situation [42]. An additional strength of this study is that the model assumes multiple screening programmes. The results of this study provide more valuable recommendations for the development of detailed screening programmes than previous studies where only repeat screening and no screening programmes were designed for comparison. There are also some unavoidable limitations to this study. First, the Markov model designed for this study does not fully model the real natural history of *H. pylori* infection and associated disease. Second, we did not model gender and region separately to discuss cost-effectiveness, and *H. pylori*-related disease varied significantly by gender and region. This study remedies this problem through uncertainty-sensitive analysis, and in terms of the results, the results of this study are sufficient to evaluate the bias introduced by these differences.

This study provides a comprehensive analysis of the cost-effectiveness of different *H. pylori* screening strategies for asymptomatic populations in China, and the results can be used as a reference for digestive disease prevention policies developed by health administrations. The generality of this study is worth discussing. The study findings should be applied to developing countries and regions similar to China. The rates of *H. pylori* infection and recurrence, the cost of screening and eradication treatment, and patient compliance vary greatly due to the actual situation in different countries and regions [17,21]. Therefore, we believe that the results of this study should be extrapolated with caution. It is worth noting that the results of this study regarding age-related trends in screening are generalizable. The same trend has been found in similar studies in other countries and regions, suggesting that early screening is more cost-effective for all populations.

## 6. Conclusions

In summary, it is cost-effective to screen asymptomatic Chinese populations for *H. pylori* and to eradicate treatment in patients with positive screening results. *H. pylori* eradication treatment has significant benefits for the prevention of stomach cancer and peptic ulcers. Therefore, from a national payer perspective, we recommend screening for *H. pylori* in asymptomatic Chinese populations and eradication treatment for those who test positive. We also recommend but not the participation rate of screening in asymptomatic people, increasing the frequency of repeat screening. From an individual payer’s perspective we also recommend screening and eradication of *H. pylori* for asymptomatic people, who can choose a screening programme they can afford according to their financial situation. High-frequency repeat screening has not been shown to improve the effectiveness of peptic ulcer and gastric cancer prevention in Chinese asymptomatic populations.

## Figures and Tables

**Figure 1 ijerph-19-09986-f001:**
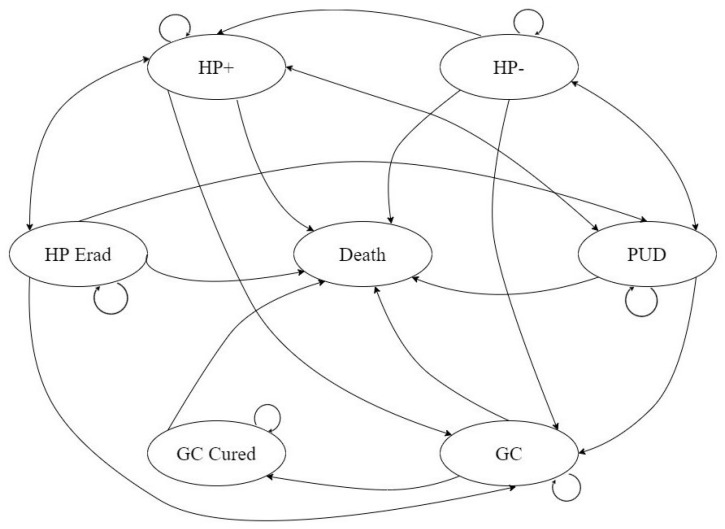
Schematic representation of the Markov model. *H. pylori* uninfected (HP−); *H. pylori* infected (HP+); *H. pylori* successfully eradicated (HP Erad); gastric cancer (GC); gastric cancer cured (GC Cured); peptic ulcer disease (NUD). At the beginning, all individuals in the cohort entered the model from two healthy states, i.e., HP− and HP+. During each Markovian cycle (1 year), individuals moved between the different health states, the.

**Figure 2 ijerph-19-09986-f002:**
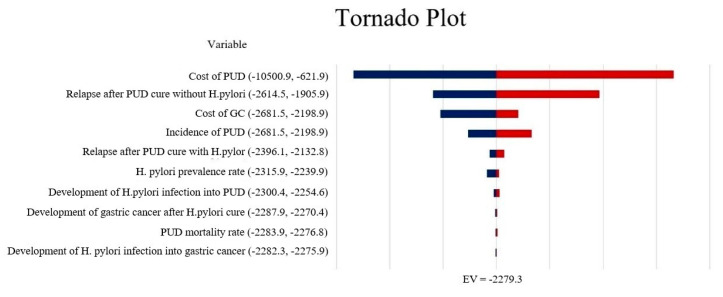
Tornado diagram showing the deterministic sensitivity analysis of the Markov model simulation.

**Figure 3 ijerph-19-09986-f003:**
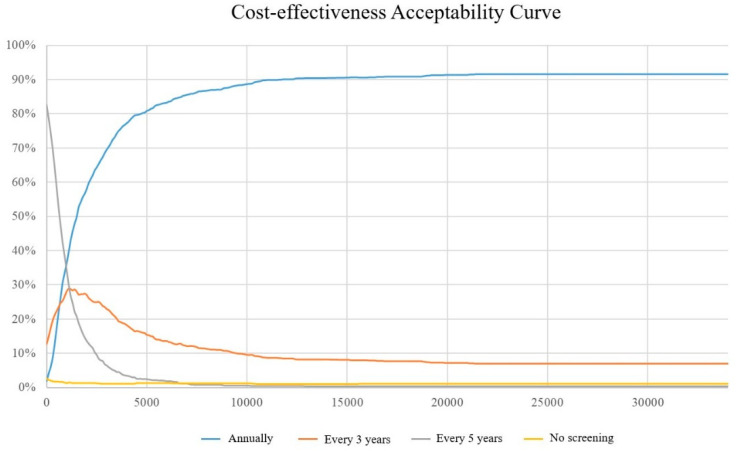
Cost-Effectiveness acceptability curve.

**Figure 4 ijerph-19-09986-f004:**
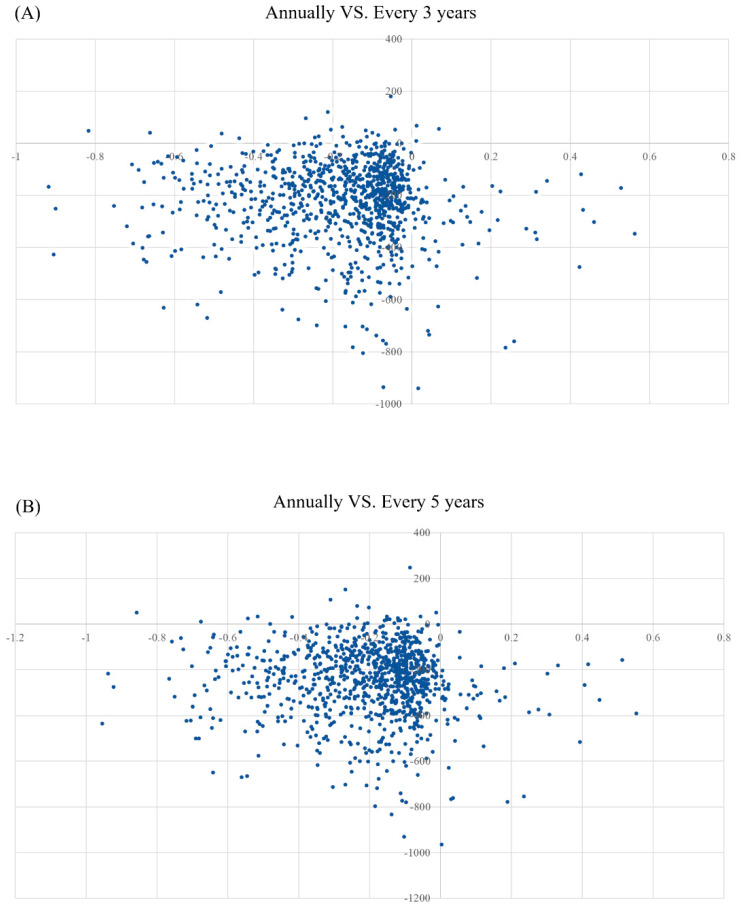
(**A**) CE plane showing the incremental costs and incremental QALYs of 1000 simulations for Annually group vs. Every 3 years group. (**B**) CE plane showing the incremental costs and incremental QALYs of 1000 simulations for Annually group vs. Every 3 years group.

**Figure 5 ijerph-19-09986-f005:**
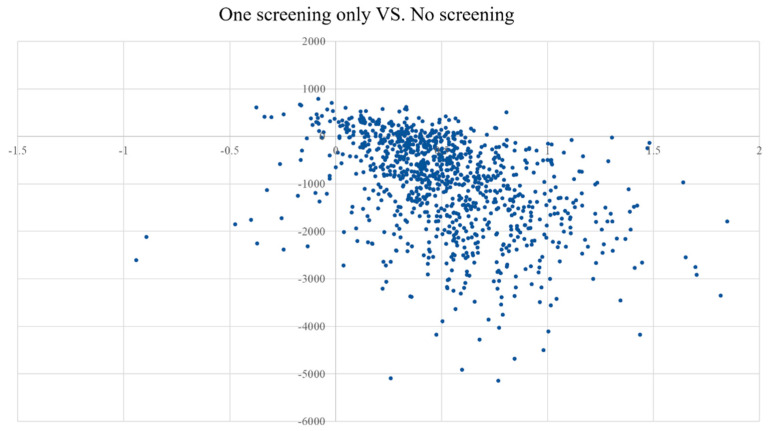
CE plane showing the incremental costs and incremental QALYs of 1000 simulations for One screening only group vs. No screening group.

**Table 1 ijerph-19-09986-t001:** Parameters in the model. PUD, peptic ulcer disease; UBT, urea breath test; NA, not applicable; Local data were obtained from reports published by the National Health and Family Planning Commission of China or the State Food and Drug Administration of China.

Variables	Baseline	Range	Distribution	References
Epidemiological parameters				
Annual *H. pylori* infection rate (%)	2.2	0.8–4	β	[17]
UBT sensitivity (%)	96	95–97	β	[18]
UBT specificity (%)	94	92–95	β	[18]
Eradication rate of bismuth quadruple therapy(%)	85.51	74.71–96.41	β	[19]
Five-year survival rate for gastric cancer (%)	47	45.0–49.5	β	[20]
Annual *H. pylori* reinfection rate after eradicated (%)	2.82	2.6–2.9	β	[21]
Development of *H. pylori* infection into gastric cancer (%)	28	18–56	β	[22]
Risk of gastric cancer development (%)	0.1186	0.099–0.14	β	[23]
Development of gastric cancer after *H. pylori* cure (%)	0.17	0.07–0.36	β	[23]
Development of PUD without *H pylori* infection (%)	9	7.2–10.8	β	[24]
Development of PUD with *H pylori* infection (%)	14	13.3–14.7	β	[25]
PUD developing into gastric cancer (%)	0.7	0.16–2	β	[26]
PUD mortality rate (%)	2.53	2.44–2.63	β	[27]
Relapse after PUD cure with *H. pylori*(%)	16.3	10.5–22.0	β	[28]
Relapse after PUD cure without *H. pylori* (%)	11.899	7.665–16.06	β	[28]
*H. pylori* infection rate (%)	50	15.5–83.4	β	[29]
Costs (US dollars)				
*H. pylori* screening test (UBT)	20.87	12.04–40.14	Gamma	Local data
*H. pylori* eradication treatment	28.27	9.78–59.23	Gamma	Local data
Average annual cost of PUD	1288.3	257.66–6441.50	Gamma	Local data
Average annual cost of Gastric cancer	3182.33	636.47–15911.65	Gamma	Local data
Health-state utility				
Health	1	NA	NA	Assumption
PUD	0.886	0.841–0.922	β	[30]
Gastric cancer	0.603	0.470–0.730	β	[30]
Cured gastric cancer	0.951	0.928–0.969	β	[30]
Death	0	NA	NA	

**Table 2 ijerph-19-09986-t002:** Distribution of gastric cancer and death states (each group had 10,000 persons).

Cycles (Years)	Gastric Cancer			Death		
	Annually	Every 3 Years	Every 5 Years	Annually	Every 3 Years	Every 5 Years
10	18	19	19	83	83	84
20	20	22	24	93	94	94
30	12	13	16	85	85	86
40	10	10	12	76	77	77
50	10	11	10	69	69	68
60	10	10	10	62	61	62

**Table 3 ijerph-19-09986-t003:** The result of base-case cost-effectiveness analysis. QALY, quality-adjusted life-year; EFF: effectiveness; ICER: incremental cost-effectiveness ration.

Screening Programme	Cost	INCR Cost	Life Year	ICER YoLS	EFF	ICER
Annually	2487.08		45.43		22.78	
Every 3 years	2266.06	−221.02	45.01	516.41	22.61	1317.48
Every 5 years	2241.19	−245.88	44.96	522.05	22.59	1277.92

**Table 4 ijerph-19-09986-t004:** The result of scenario sensitivity analyses. QALY, quality-adjusted life-year; EFF: effectiveness; ICER: incremental cost-effectiveness ration.

Screening Age	Screening Programme	Cost	INCR Cost	EFF	INCR EFF	ICER
20	Annually	2441.80		22.78		
	Every 3 years	2223.98	−217.82	22.61	−0.18	1238.48
	Every 5 years	2204.59	−237.21	22.58	−0.20	1163.71
30	Annually	2406.57		21.65		
	Every 3 years	2184.48	−222.09	21.5	−0.15	1480.6
	Every 5 years	2159.73	−246.84	21.48	−0.17	1452
40	Annually	2333.99		19.97		
	Every 3 years	2131.73	−202.26	19.85	−0.12	1685.5
	Every 5 years	2106.61	−227.38	19.83	−0.14	1624.14
50	Annually	2102.4		17.46		
	Every 3 years	1916.41	−185.99	17.37	−0.09	2066.56
	Every 5 years	1895.36	−207.04	17.36	−0.1	2070.4
60	Annually	1692.51		13.74		
	Every 3 years	1538.08	−154.43	13.69	−0.05	3088.6
	Every 5 years	1518.19	−174.32	13.68	−0.06	2905.33
70	Annually	1084.71		8.21		
	Every 3 years	992.76	−91.95	8.19	−0.02	4597.5
	Every 5 years	970.41	−114.3	8.18	−0.03	3810
No screening		3617.40		13.67		
One screening only		1591.63	−2025.77	22.22	8.54	−237.07

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
