# Peer review of "Cost-Effectiveness Analysis of the Helicobacter Pylori Screening Programme in an Asymptomatic Population in China"

_ijerph, 2022, doi:10.3390/ijerph19169986_

Round 1

Reviewer 1 Report

Authors Feng et al. present a cost-effectiveness analysis of H. Pylori screening in an asymptomatic Chinese population. The authors have used Bayesian inference to calculate significance of QALYs per YoLS to assess direct healthcare costs of yearly, three-year, and five-year screening programs for asymptomatic individuals. They have also assessed this relationship with respect to age. This is an interesting manuscript and is generally well-written. Given the importance of H. Pylori treatment in China, this is a valuable manuscript that will be of interest to readers. Nevertheless, there are a few corrections that should be made to improve the paper.

Major concern:

1.     While the authors have identified several key limitations of their study, they have not discussed the generalizability of their findings. What is the translational potential of this work? Can the main findings of screening frequency apply to non-Chinese populations? Can the identified age-related trends be reasonably generalized to other populations? Are there other populations with similar relevant characteristics to which these results can apply? Please address this critical limitation and the generalizability of your findings in the strengths and limitations section.

Minor concerns:

1.     Italicize all instances of H. Pylori.

2.     Alphabetize the keywords listed on the title page.

3.     Lines 50-51: this sentence does not read well. Please just end the sentence with ‘in previous Markov model studies.’.

4.     Define abbreviations contained in Figure 1 in the caption.

5.     Line 82: capitalize ‘parameters’.

6.     Table 1: it is not ‘gama’ but rather ‘gamma’. Please change all instances of the former to the latter.

7.     Cite the studies referenced in the sentence on lines 93-94.

8.     Number all equations (i.e., Equation 1, Equation 2, etc.) and refer to them as such.

9.     Cite the studies referenced in the sentence on line 104-106.

10.  Line 107: cite the 2020 Chinese Health Statistics.  

11.  The authors have mistakenly referred to Figure 3 as a ‘cost-effectiveness acceptable curve’ both in the text as well as the Figure 3 caption. The correct name is ‘cost-effectiveness acceptability curve’. Please correct.

12.  Lines 151-153: no need to capitalize the names of these analyses. Lowercase is fine, only capitalize their abbreviations.

13.  Change ‘spent’ to ‘survived’ on lines 163 and 165.

14.  Line 177: fix “parameteron’.

15.  Numbers >10 do not need to be spelled out and should instead be represented numerically (see Figure 4 and Figure 5 captions, check elsewhere as well).

16.  Place table 4 on a new page, it is poorly represented being split between pages.

17.  Cite the referenced studies in the sentence contained on lines 273-274.

18.  Line 282: do not capitalize ‘Labour’.

19.  Line 287: delete ‘the’ from “In the light of’.

20.  Line 328: replace ‘increasing’ with ‘but not’, and delete ‘is not effective’ from the end of the sentence to enhance its readability.

21.  Bold and fix the manuscript subsections at the end of the study (i.e., declaration, author contributions, acknowledgements, etc.). It is not clear as-is.

I therefore recommend the paper be published following successful amelioration of the above-listed concerns. I would be happy to receive a revised manuscript for further consideration.

Author Response

Response to Reviewer 1 Comments

Thanks to your comments, we have revised the manuscript in response to these suggestions. We hope that this revision is to your satisfaction.

Point 1:

Major concern:

  1. While the authors have identified several key limitations of their study, they have not discussed the generalizability of their findings. What is the translational potential of this work? Can the main findings of screening frequency apply to non-Chinese populations? Can the identified age-related trends be reasonably generalized to other populations? Are there other populations with similar relevant characteristics to which these results can apply? Please address this critical limitation and the generalizability of your findings in the strengths and limitations section.

Response1:

We add a discussion of the generalizability of this study in the strengths and limitations section. (lines 327-337) We believe that H. pylori screening is cost-effective for the general population in China but that caution is needed when extrapolating this study to other countries and regions.

Point 2:

Italicize all instances of H. Pylori.

Response 2:

We have changed ‘H. pylori’ to italics throughout the text.

Point 3:

Alphabetize the keywords listed on the title page.

Response 3:

Corrections have been made to the keyword sorting.

Point 4:

Lines 50-51: this sentence does not read well. Please just end the sentence with ‘in previous Markov model studies.’.

Response 4:

This has been corrected.

Point 5:

Define abbreviations contained in Figure 1 in the caption.

Response 5:

We have added an introduction to the model structure in Figure 1 and a definition of abbreviations.

Point 6:

Line 82: capitalize ‘parameters’.

Response 6:

This has been corrected.

Point 7:

Table 1: it is not ‘gama’ but rather ‘gamma’. Please change all instances of the former to the latter.

Response 7:

This has been corrected.

Point 8:

Cite the studies referenced in the sentence on lines 93-94.

Response 8:

We present a more accurate presentation of the conversion method for transfer probabilities, while citing more accurate literature. (Lines 99-101) The added references are serial numbers 31 and 32.

Point 9:

Number all equations (i.e., Equation 1, Equation 2, etc.) and refer to them as such.

Response 9:

The number of the equation has been added here.

Point 10:

Cite the studies referenced in the sentence on line 104-106.

Response 10:

References have been added, numbered 19. (lines 109-111)

Point 11:

Line 107: cite the 2020 Chinese Health Statistics. 

Response 11:

References have been added, numbered 35. (lines 114-115)

Point 12:

The authors have mistakenly referred to Figure 3 as a ‘cost-effectiveness acceptable curve’ both in the text as well as the Figure 3 caption. The correct name is ‘cost-effectiveness acceptability curve’. Please correct.

Response 12:

This has been corrected.

Point 13:

Lines 151-153: no need to capitalize the names of these analyses. Lowercase is fine, only capitalize their abbreviations.

Response 13:

Abbreviation formatting in the full text has been corrected.

Point 14:

Change ‘spent’ to ‘survived’ on lines 163 and 165.

Response 14:

This has been corrected.

Point 15:

Line 177: fix “parameteron’.

Response 15:

Here 'parameteron' has been amended to 'parameter on'.

Point 16:

Numbers >10 do not need to be spelled out and should instead be represented numerically (see Figure 4 and Figure 5 captions, check elsewhere as well).

Response 16:

This has been modified and checked in other locations throughout the text.

Point 17:

Place table 4 on a new page, it is poorly represented being split between pages.

Response 17:

Here is the result of the magazine's typesetting, and we spoke with the editor about it here.

Point 18:

Cite the referenced studies in the sentence contained on lines 273-274.

Response 18:

References have been added, numbered 42. (lines 278-279)

Point 19:

Line 282: do not capitalize ‘Labour’.

Response 19:

This has been corrected.

Point 20:

Line 287: delete ‘the’ from “In the light of’.

Response 20:

This has been corrected.

Point 21:

Line 328: replace ‘increasing’ with ‘but not’, and delete ‘is not effective’ from the end of the sentence to enhance its readability.

Response 21:

This has been corrected.

Point 22:

Bold and fix the manuscript subsections at the end of the study (i.e., declaration, author contributions, acknowledgements, etc.). It is not clear as-is.

Response 22:

We have bolded the subheadings of these sections.

Reviewer 2 Report

Dear authors.

I have the following comments:

1.   This paper provides a clear description of the background, the prevalence of H. pylori infection, and the current problem in dealing with H. pylori infection in China. It is socially significant.

2.   The use of a Markov decision model is a suitable solution for the cost-effectiveness issue.

3.   Model specification is generally clear. However, more illustration of the model is needed. For example, the symbols used in Figure 1 can be immediately explained after the figure, which makes it easier for the reader to read. “S is the rate,”(line 99) of what? T for what (line 98)?

4.   The major concern is the data description. a) The sources of data need further discussion and details. For example, “Annual H pylori infection rate (%), 0.01, Range: 0.25-4… Reference: 17”. Reference 17 is a study on Singapore, not the Chinese data… The sources of “Local Data” in Table 1. b). The scales of the data are not consistent. For example, 0.01 cannot be in the range 0.25-4. Because of the issues in data, the results derived from these data may not be reliable.

5.   Many small problems in English grammar, expressions, and terms. But these problems can be fixed easily.

Author Response

Response to Reviewer 2 Comments

Thank you for your suggestions and comments, which have played an important role in improving the quality of our manuscript. We have revised the manuscript in accordance with your suggestions, and some of the results have been recalculated to ensure that they are accurate and reliable.

Point 1:

Model specification is generally clear. However, more illustration of the model is needed. For example, the symbols used in Figure 1 can be immediately explained after the figure, which makes it easier for the reader to read. “S is the rate,”(line 99) of what? T for what (line 98)?

Response 1:

Thank you very much for your suggestion. We add a description of the model structure in Figure 1 and how the model is run. We provide a more detailed description of the transformation method for transfer probabilities. (Lines 99-105)

Point 2:

The major concern is the data description. a) The sources of data need further discussion and details. For example, “Annual H pylori infection rate (%), 0.01, Range: 0.25-4. Reference: 17”. Reference 17 is a study on Singapore, not the Chinese data… The sources of “Local Data” in Table 1. b). The scales of the data are not consistent. For example, 0.01 cannot be in the range 0.25-4. Because of the issues in data, the results derived from these data may not be reliable.

Response 2:

Thank you very much for your suggestion. There was a clerical error in the transcription and therefore an inconsistency in the scale of the data, which we have corrected in this revision.

In addition, we have replaced the infection rate data with data from a more reliable study in this revision. This data was determined to be from a Chinese study. Since we recalculated the results due to the replacement of data, the new results did not differ significantly from the previous results, and therefore the conclusions of this study did not change significantly due to this change.

Finally, we provide more detail on the source of the local data in the notes to Table 1. (Lines 86-88)

Round 2

Reviewer 2 Report

Glad to see the changes and corrections. This version is good.